# The global distribution and spread of the mobilized colistin resistance gene *mcr-1*

Ruobing Wang[1], Lucy van Dorp [2], Liam P. Shaw [2], Phelim Bradley[3], Qi Wang[1], Xiaojuan Wang[1], Longyang Jin[1], Qing Zhang[4], Yuqing Liu[4], Adrien Rieux[5], Thamarai Dorai-Schneiders[6], Lucy Anne Weinert[7], Zamin Iqbal [3,8], Xavier Didelot[9], Hui Wang[1] & Francois Balloux [2]

Colistin represents one of the few available drugs for treating infections caused by carbapenem-resistant *Enterobacteriaceae*. As such, the recent plasmid-mediated spread of the colistin resistance gene *mcr-1* poses a significant public health threat, requiring global monitoring and surveillance. Here, we characterize the global distribution of *mcr-1* using a data set of 457 *mcr-1*-positive sequenced isolates. We find *mcr-1* in various plasmid types but identify an immediate background common to all *mcr-1* sequences. Our analyses establish that all *mcr-1* elements in circulation descend from the same initial mobilization of *mcr-1* by an IS*Apl1* transposon in the mid 2000s (2002–2008; 95% highest posterior density), followed by a marked demographic expansion, which led to its current global distribution. Our results provide the first systematic phylogenetic analysis of the origin and spread of *mcr-1*, and emphasize the importance of understanding the movement of antibiotic resistance genes across multiple levels of genomic organization.

[1] Department of Clinical Laboratory, Peking University People's Hospital, Beijing 100044, China. [2] UCL Genetics Institute, University College London, Gower Street, London WC1E 6BT, UK. [3] Wellcome Trust Centre for Human Genetics, University of Oxford, Oxford OX3 7BN, UK. [4] Institute of Animal Science and Veterinary Medicine, Shandong Academy of Agricultural Sciences, Shandong Province, Jinan 250100, China. [5] UMR PVBMT, CIRAD, 97410 St Pierre, Reunion, France. [6] Division of Infection and Pathway Medicine, 49 Little France Crescent, Edinburgh EH16 4SB, UK. [7] Department of Veterinary Medicine, Cambridge CB3 0ES, UK. [8] European Molecular Biology Laboratory, European Bioinformatics Institute (EMBL-EBI), Wellcome Genome Campus, Cambridge CB10 1SD, UK. [9] Department of Infectious Disease Epidemiology, Imperial College London, Norfolk Place 21, London W2 1PG, UK. These authors contributed equally: Ruobing Wang, Lucy van Dorp, Liam P. Shaw. Correspondence and requests for materials should be addressed to H.W. (email: wanghui@pkuph.edu.cn) or to F.B. (email: f.balloux@ucl.ac.uk)

Colistin was largely abandoned as a treatment for bacterial infections in the 1970s owing to its toxicity and low renal clearance, but has been reintroduced in recent years as an antibiotic of 'last resort' against multi-drug-resistant infections[1]. It is therefore alarming that the prevalence of resistance to colistin has become a significant concern, following the identification of plasmid-mediated colistin resistance conferred by the *mcr-1* gene in late 2015[2]. Colistin resistance is emblematic of the growing problems of antimicrobial resistance worldwide.

Up until 2015, resistance to colistin had only been linked to mutational and regulatory changes mediated by chromosomal genes[3,4]. The mobilized colistin gene *mcr-1* was first described in a plasmid carried by an *Escherichia coli* isolated in China in April 2011[2]. The presence of colistin resistance on mobile genetic elements poses a significant public health risk, as these can spread rapidly by horizontal transfer, and may entail a lower fitness cost[5]. At the time of writing, *mcr-1* has been identified in numerous countries across five continents. Significantly, *mcr-1* has also been observed on plasmids containing other anti-microbial resistance genes such as carbapenemases[6–8] and extended-spectrum β-lactamases[9–11].

The *mcr-1* element has been characterized in a variety of genomic backgrounds, consistent with the gene being mobilized by a transposon[12–16]. To date, *mcr-1* has been observed on a wide variety of plasmid types, including IncI2, IncHI2, and IncX4[17]. Intensive screening efforts for *mcr-1* have found it to be highly prevalent in a number of environmental settings, including the Haihe River in China[18], recreational water at public urban beaches in Brazil[19], and fecal samples from otherwise healthy individuals[20,21]. Although both Brazil and China have now banned the use of colistin in agriculture, the evidence that *mcr-1* can spread within hospital environments even in the absence of colistin use[22] as well as in the community[21] raises the possibility that the spread of *mcr-1* will not be contained by these bans.

The global distribution of *mcr-1* over at least five continents is well documented, but little is known about its origin, acquisition, emergence, and spread. In this study, we aim to shed light on these fundamental issues using whole-genome sequencing (WGS) data from 110 novel *mcr-1*-positive isolates from China in conjunction with an extensive collection of publicly available sequence data sourced from the NCBI repository as well as the Short Read Archive (SRA).

Our data and analyses support an initial single mobilization event of *mcr-1* by an IS*Apl1-mcr-1-orf-*IS*Apl1* transposon around 2006. The transposon was immobilized on several plasmid backgrounds following the loss of the flanking IS*Apl1* elements, and spread through plasmid transfer. The current distribution of *mcr-1* points to a possible origin in Chinese livestock. Our results illustrate the complex dynamics of antibiotic resistance genes across multiple embedded genetic levels (transposons, plasmids, bacterial lineages and bacterial species), previously described as a nested 'Russian doll' model of genetic mobility[23].

## Results

**Data set.** We compiled a global data set of 457 *mcr-1*-positive isolates (Fig. 1a), including 110 new WGS from China, of which 107 were sequenced with Illumina short reads and three with PacBio long-read technology. One hundred and ninety-five isolates were sourced from publicly available assemblies in the NCBI GenBank repository (73 completed plasmids, 1 complete chromosome, 121 assemblies). A further 152 sequences were sourced from the NCBI SRA, after being identified as *mcr-1*-positive using a k-mer index of a snapshot of the SRA as of December 2016 (see Methods). The whole data set consists of 256 short-read data sets, 6 long-read PacBio WGS, 121 draft assemblies, and 74 completed

assemblies. Accession numbers and metadata for the 457 isolates are provided in supplementary data 1.

Isolates carrying *mcr-1* were identified from 31 countries (Fig. 1a). The countries with the largest numbers of *mcr-1*-positive samples are China (212), Vietnam (58) and Germany (25). Within China, nearly half (45%) of positive isolates stem from the Shandong province (Fig. 1b). The vast majority of *mcr-1*-positive isolates belong to *E. coli* (411), but the data set also comprises *mcr-1*-positive isolates from another seven bacterial species: *Salmonella enterica* (29), *Klebsiella pneumoniae* (8), *Escherichia fergusonii* (2), *Kluyvera ascorbata* (2), *Citrobacter braakii* (2), *Cronobacter sakazakii* (1), and *Klebsiella aerogenes* (1) (Fig. 1a). The majority of isolates for which sampling dates were available (80%), were collected between 2012 and 2016, with the oldest available isolates dating back to 2008 (Fig. 1c). Isolates with metadata on the sample source (*n* = 360) came from a range of animal (*n* = 222), human (*n* = 108) and environmental (*n* = 30) hosts.

The large number of *mcr-1*-positive isolates from China, and the high incidence in the Shandong province can be largely ascribed to the inclusion of our 110 newly sequenced isolates including 49 from Shandong and to another 37 isolates from a previous large sequencing effort[13]. However, even after discounting the isolates from these two sources, China remains, together with Vietnam one of the two countries with the highest number of sequenced *mcr-1*-positive isolates.

**Evolutionary model.** It has been proposed that *mcr-1* is mobilized by a composite transposon formed of a ~2600 bp region containing *mcr-1* (1626 bp) and a putative open reading frame encoding a PAP2 superfamily protein (765 bp), flanked by two IS*Apl1* insertion sequences[12]. IS*Apl1* is a member of the IS30 family of insertion sequences, which utilize a 'copy-out, paste-in' mechanism with a targeted transposition pathway requiring the formation of a synaptic complex between an inverted repeat (IR) in the transposon circle and an IR-like sequence in the target. Snesrud and colleagues[12] hypothesized that after the initial formation of such a composite transposon, these insertion sequences would have been lost over time, leading to the stabilization of *mcr-1* in a diverse range of plasmid backgrounds (Fig. 2). In the following, we sought to test this model by performing an explicit phylogenetic analysis of the region surrounding *mcr-1* using our comprehensive global data set.

**Immediate genomic background of *mcr-1*.** If there had been a unique formation event for the composite transposon, followed by progressive transposition and loss of insertion sequences, we would expect to be able to identify a common immediate background region for *mcr-1* in all samples. Indeed, we were able to identify and align a shared region or remnants of it in all 457 sequences surrounding *mcr-1* (see Methods), supporting a single common origin for all *mcr-1* elements sequenced to date (Fig. 3a). The majority of the sequences contained no trace of IS*Apl1* (*n* = 260) indicating that the *mcr-1* transposon had been completely stabilized in their genomic background. Forty-two sequences contained indication of the presence of IS*Apl1* both upstream and downstream, either in full copies (*n* = 16), a full copy upstream and a partial copy downstream (*n* = 7), a partial copy upstream and a full copy downstream (*n* = 1), or partial copies upstream and downstream (*n* = 18). Some sequences only had IS*Apl1* present upstream as a complete (*n* = 55) or partial (*n* = 99) sequence, and one sequence had only a partial downstream IS*Apl1* element. The downstream copy of IS*Apl1* was inverted in some sequences (*n* = 3) and some sequences had full

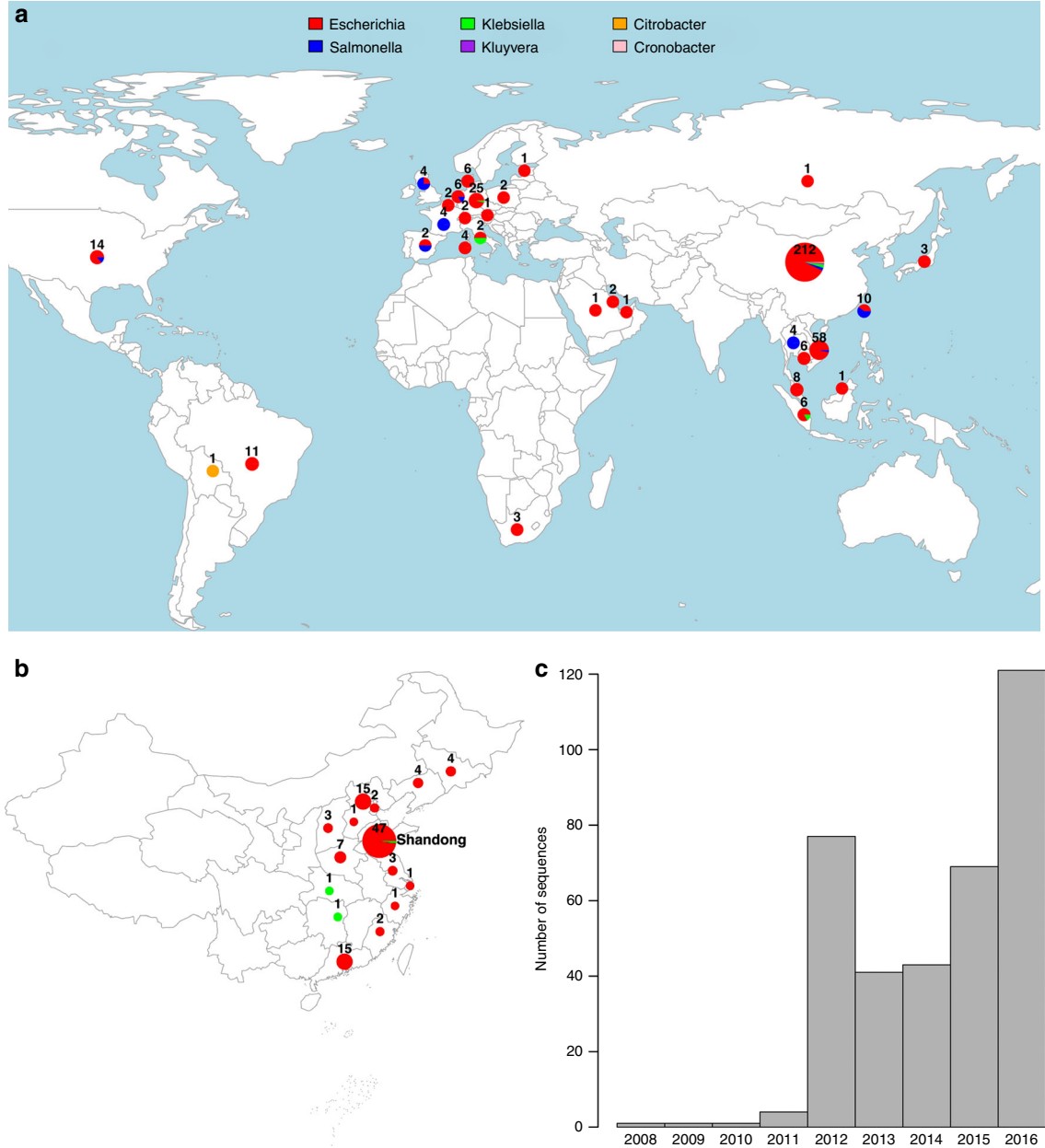

**Fig. 1** Overview of the *mcr-1*-positive isolates included. **a** Global map of *mcr-1*-positive isolates included colored by genus with the number and size of pies providing the sample size per location; **b** Map of novel Chinese isolates sequenced for this study; **c** Histogram of sampling dates (years) of the isolates. Maps were created using the R package *rworldmap* using the public domain Natural Earth data set

copies of IS*Apl1* present elsewhere on the same contig ($n = 7$), consistent with its high observed activity in transposition[24].

Further inspection of the transposon alignment revealed that the 186 bp region between the 3'-end of the upstream IS*Apl1* and *mcr-1* contained IR-like sequences similar to the IRR and IRL of IS*Apl1* (respectively: 93–142 bp, 23/50 identity; and 125–175 bp, 21/50 identity). The most variable positions in this 186 bp region were at 177 bp and 142 bp, approximately coinciding with the end of the alignment with the IRs and were more variable in sequences lacking IS*Apl1*, suggesting possible loss of function of the transposition pathway associated with IS*Apl1* (Fig. 3d). Some of these SNPs occurred in a stretch previously identified as the promoter region for *mcr-1*[25], and this region showed strong signals of recombination. A small number of sequences (3%) had SNPs present in *mcr-1* itself. These tended to be at the upstream/

5'-end of the sequence, particularly in the first three positions. A subset of the sequences from Vietnam ($n = 28$) included a secondary 1.7 kb insertion downstream of *mcr-1* containing a putative transposase, indicating subsequent rearrangements involving this region after initial mobilization of the transposon (Fig. 3e).

To reconstruct the phylogenetic history of the composite *mcr-1* transposon, we created a sequence alignment for 457 sequences (Fig. 3c) after removal of recombinant regions identified with ClonalFrameML, including the region immediately upstream of *mcr-1* between positions 1212–1247 (Fig. 3d). A midpoint-rooted maximum parsimony phylogeny showed that there was a dominant sequence type with subsequent diversification, likely indicating the ancestral form of the composite transposon (Fig. 4). There was no discernible clustering of isolates by bacterial species

(Supplementary Fig. 1) or sample source (Supplementary Fig. 2), suggesting the composite transposon does not evolve differently in these different backgrounds.

A Bayesian dating approach (BEAST) was applied to infer a timed phylogeny of the maximal alignable region of the *mcr-1* carrying transposon (see Methods). Based on this 3522 site alignment we infer a common ancestor for 364 dated isolates in 2006 (Supplementary Fig. 3; 2002–2008 95% highest poster density (HPD) with a strict clock and coalescent model) with a mutation rate around $7.51 \times 10^{-5}$ substitutions per site per year (Supplementary Table 1). There was no clear overall geographic clustering in the Maximum Clade Credibility (MCC) tree (Supplementary Figure 4).

**Wider genomic background of *mcr-1*.** Next, we explored the wider genomic background upstream and downstream of the conserved transposon sequences. We had sufficiently long assembled contigs for 182 isolates to identify plasmid types based on co-occurrence with plasmid replicons (see Methods) and identified *mcr-1* in 13 different plasmid backgrounds. IncI2 and IncX4 were the dominant plasmid types, accounting for 51 and 38% of the isolates, respectively (Fig. 5) similar to the proportions observed by Matamoros et al.[15] One isolate in our data set was definitively located on a complete chromosome. Though, we cannot rule out the presence of a few other chromosomal copies of *mcr-1* located on short contigs.

The distribution of transposons carrying one or two copies of ISApl1 was highly heterogeneous across these plasmid types. For example, sequences with one or two copies of ISApl1 were found on six and four types, respectively, which supports their mobility compared with those without ISApl1, which were found in five plasmid types. Of the contigs carrying one copy of ISApl1, 61% were found in IncI2 plasmids, and 50% of contigs carrying two copies of ISApl1 belonged to IncHI2 plasmids. Conversely, the common IncX4 plasmids carried only two transposons with two copies of ISApl1 and none with a single copy of the element.

We identified two extended plasmid backbone sequences that could be aligned. The first such alignment encompasses a shared sequence of 7161 bp between 108 plasmid backgrounds and has been previously referred to as 'Type A'[13]. These sequences contain 54 sequences co-occurring with an IncI2 replicon, with 54 of unknown plasmid type, and encompass a large fraction of the genetic diversity found in the *mcr-1* transposon, although a large proportion (9/108) belonged to the dominant sequence type (i.e., B07_1_WFA61 in Fig. 4). The second alignment is 34,761 bp long and is common to nine IncX4 plasmids and partly overlaps with a background previously defined as 'Type D'[13].

We applied BEAST to infer a timed phylogeny for each of these alignable regions after removal of SNPs showing evidence of recombination. For the IncI2 background we infer that a common ancestor to all 108 isolates existed in 2006 (1998–2010 95% CI relaxed exponential clock model) assuming a constant population size model (Supplementary Fig. 5). For the IncX4 backgrounds we dated the common ancestor of the eight isolates to 2011 (2010–2013 95% CI relaxed exponential clock model) assuming a constant population size model (Supplementary Fig. 6). Posterior density distributions of root dating under different population models are shown in Supplementary Figs. 7–8. The difference in dating inferred for these two plasmid backgrounds and the recent date obtained for IncX4 highlight the dynamic nature of the integration of the *mcr-1* carrying transposon, even if in the IncX4 phylogeny isolates from East Asia and Europe and the Americas cluster together. The inferred mutation rates obtained for the IncI2 and IncX4 backgrounds consistently lie around $5$–$10 \times 10^{-5}$ substitutions per site per year (Supplementary Table 1).

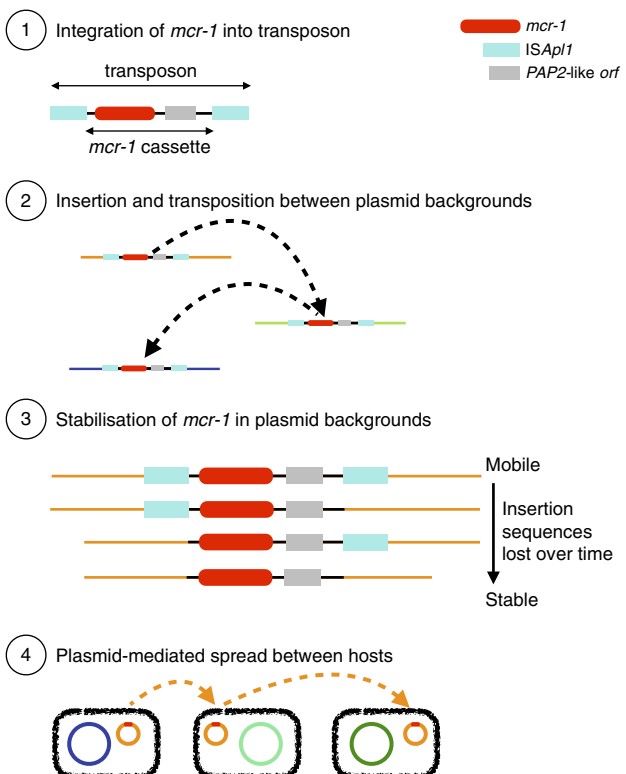

**Fig. 2** Schematic representation of the evolutionary model for the steps in the spread of the *mcr-1* gene. (1) The formation of the original composite transposon, followed by (2) transposition between plasmid backgrounds and (3) stabilization via loss of ISApl1 elements before (4) plasmid-mediated spread

**Environmental distribution of the composite *mcr-1* transposon.** It has been suggested that agricultural use of colistin, as has been widespread in China since the early 1980s[21], caused the initial emergence and spread of *mcr-1*[26,27]. According to the evolutionary model in Fig. 2, the ancestral mobilizable state is represented by the transposon carrying both its ISApl1 elements. The transposon is thought to lose its capability for mobilization after the loss of both ISApl1 elements, although a single copy is reportedly sufficient to keep some ability to mobilize, with the upstream copy being functionally more important[12]. Comparing human ($n = 108$) and non-human ($n = 252$) isolates, there were significantly more sequences with some trace of the insertion sequence ISApl1 both upstream and downstream in non-human isolates (32/220 vs. 5/108, $\chi^2$-test, $p = 0.033$). This comparison held when only comparing agricultural isolates to human isolates ($n = 213$) (28/213 vs. 5/108, $\chi^2$-test, $p = 0.029$). Furthermore, of the 42 isolates that had ISApl1 fragments both upstream and downstream, the majority were from Asia ($n = 30$) with only a quarter from Europe ($n = 10$) ($\chi^2$ test, $p = 0.12$). This result is not driven by an over representation of agricultural isolates from Asia in our data set ($\chi^2$ test, $p = 0.38$).

## Discussion

We assembled a global data set of 457 *mcr-1*-positive sequenced isolates and could show that there was a single integration event of *mcr-1* into an ISApl1 composite transposon, followed by its subsequent spread between multiple genomic backgrounds. Our phylogenetic analyses point to a date for the insertion of *mcr-1* into the gene transposon shared across our isolates in the mid 2000s (2002–2008 95% HPD). We could identify the likely sequence of the ancestral transposon type and show the pattern of

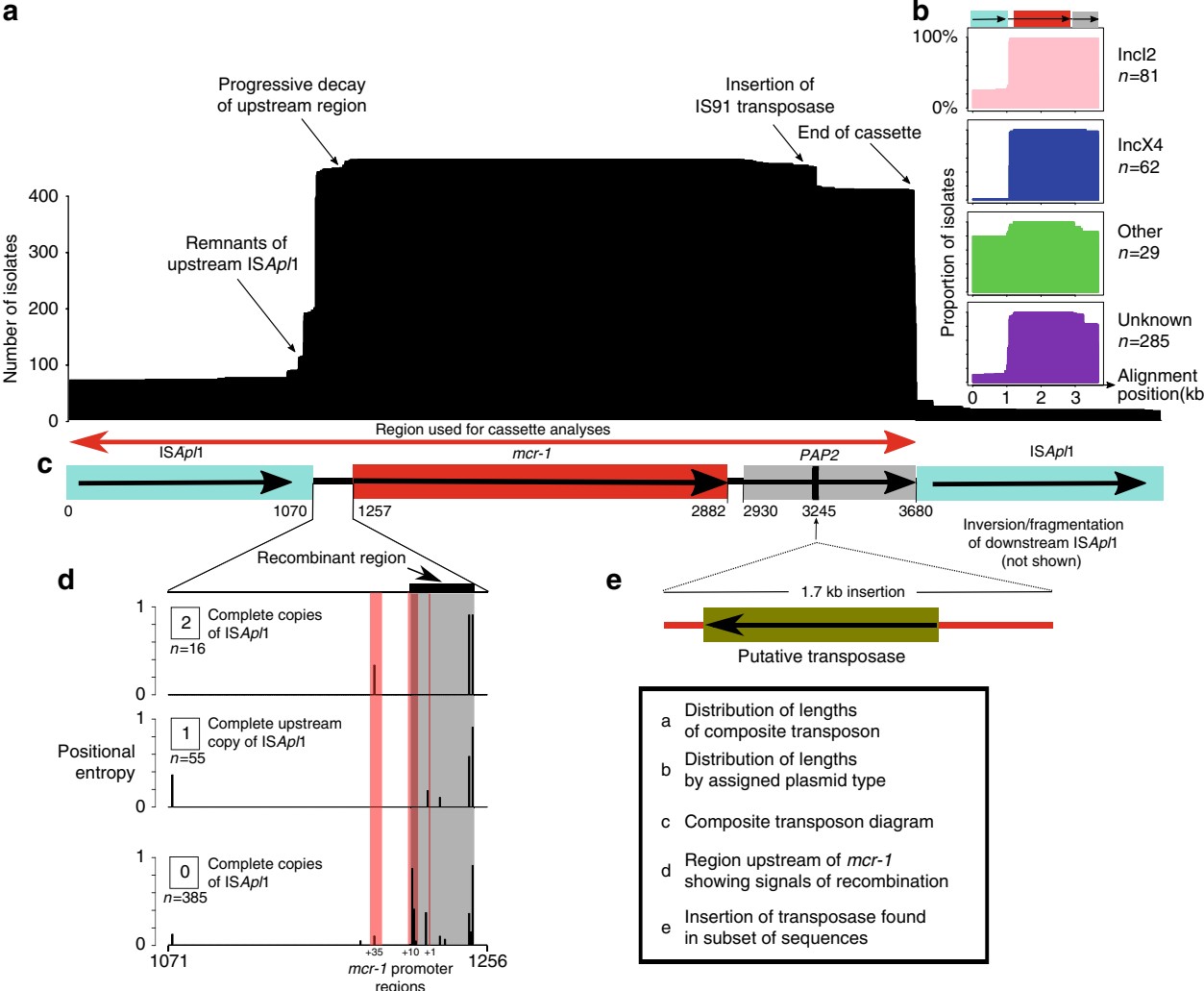

**Fig. 3** The genetic element carrying *mcr-1* is a composite transposon and is alignable across our global data set. **a** Length distribution of the alignment across sequences. **b** Length distribution subset by plasmid type. **c** The composite transposon, consisting of IS*Apl*1, *mcr-1*, a PAP2 *orf*, and IS*Apl*1. The region indicated by the red arrow was used in phylogenetic analyses, after the removal of recombination. **d** The 186 bp region upstream of *mcr-1* showed strong signals of recombination (gray box) that coincided with the promotor regions of *mcr-1* (red box), and this diverse region was removed from the subsequent alignment. **e** Twenty-eight sequences from Vietnam had a 1.7 kb insertion containing a putative transpose, suggesting subsequent rearrangement after initial mobilization

diversity supports a single mobilization with subsequent diversification during global spread.

Despite the limited number of WGSs for samples before 2012, with the oldest sequence available from 2008 (Fig. 1c), our estimate is consistent with the majority of available evidence from retrospective surveillance data,[26] which has found the presence of *mcr-1* in samples dating back to 2005 in Europe[28]. One retrospective study of Chinese isolates from 1970–2014 reported three *mcr-1*-positive *E. coli* dating from the 1980s[29], although *mcr-1* then did not reappear until 2004. This observation seems surprising in light of our results, which clearly exclude such an early spread of *mcr-1* at least on this IS*Apl*1 transposon background.

Our estimates of the age of spread of the representative IncI2 and IncX4 plasmid backgrounds are more recent, dating to ~ 2008 and 2013, respectively, but are both consistent with the age of the transposon mobilization event. We did not constrain the evolutionary rates in any of our phylogenetic analyses. It is thus encouraging that the different rates are highly consistent between the *mcr-1* transposon and the two plasmid backgrounds. Although this points to high internal consistency between our

estimates, we were surprisingly unable to find any previously published estimates for the evolutionary rate of bacterial plasmids.

The current distribution and observed genetic patterns are in line with a center of origin in China. This is the place where we observe the highest proportion of isolates carrying intact or partial copies of the IS*Apl*1 flanking elements. Transposon sequences carrying IS*Apl*1 elements were also overrepresented in environmental and agriculture isolates, relative to those collected from humans. This pattern is in line with agricultural settings acting as the source of *mcr-1* within bacteria isolated from humans[21]. The current global distribution has been achieved through multiple translocations, and is illustrated by the interspersed geographic origins in our phylogenetic reconstructions. A likely driver for the global spread is trade, in particular food animals[30] and meat, although direct global movement by colonized or infected humans[20] is also likely to have played a role in the current distribution.

The origin of *mcr-1* prior to its mobilization remains elusive. Despite an exhaustive search of sequence repositories, including

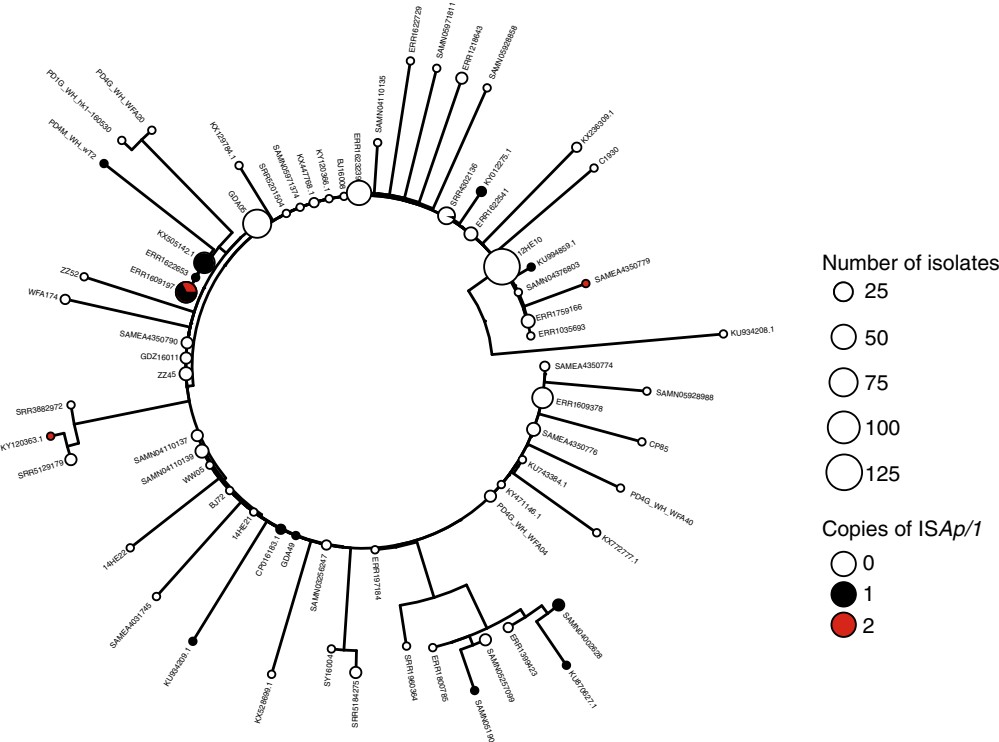

**Fig. 4** Phylogeny of the *mcr-1* composite transposon indicates a dominant sequence type with subsequent diversification. Midpoint-rooted maximum parsimony phylogeny based on the 3522 bp alignment of 457 sequences (recombinant regions removed). Size of points indicates the number of identical sequences, with a representative sequence for each shown next to each tip

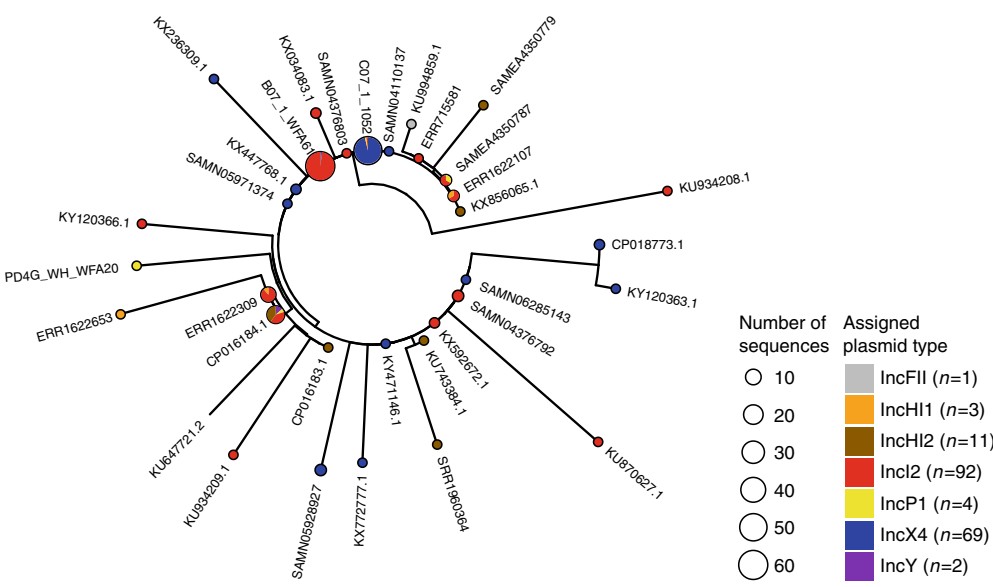

**Fig. 5** The distribution of plasmid types shown on the transposon phylogeny. Maximum parsimony tree (homoplastic sites removed, midpoint rooted, as in Fig. 4) based on the composite transposon alignment for 172 sequences containing a plasmid replicon on the same contig i.e. those with an assigned plasmid type (color). IncI2 and IncX4 are the most common plasmid types. An example sequence ID is shown for each unique sequence

the short-read archive, we found not a single *mcr-1* sequence outside the IS*Apl1* transposon background. IS*Apl1* was first identified in the pig pathogen *Actinobacillus pleuropneumoniae*[31] suggesting that it may also have been an ancestral host for *mcr-1*, although to our knowledge no *mcr-1*-positive *A. pleuropneumoniae* isolates have been described. The phosphoethanolamine transferase from *Paenibacillus sophorae* has

also been proposed as a possible candidate[32]. However, this seems most unlikely as *Paenibacili* are Gram-positive and are thus intrinsically resistant to polymixins[33]. Moreover, although the two sequences share functional similarities, this should be interpreted as a case of possible parallel evolution rather than direct filiation[33]. *Moraxella* has also been suggested as being the source of *mcr-1*[34], following the identification of genes in *Moraxella* with

limited homology to *mcr-1* (~60% nucleotide sequence identity). However, this sequence identity is too low for *Moraxella* to be considered as viable candidates for the origin of *mcr-1*. The search for the initial source of *mcr-1* remains open until a *mcr-1* sequence is identified outside of the IS*Apl1* sequence background.

We note that there are an increasing number of mobilized genes that can confer colistin resistance, with *mcr-2* reported less than a year after *mcr-1* was initially described[35] and more recent descriptions of the phylogenetically distant *mcr-3*, *mcr-4*, and *mcr-5*[36–38]. There appear to be commonalities between the mechanisms of the *mcr* genes, despite their different sequences and locations near to different insertion sequences. For example, *mcr-2* has 76.7% nucleotide identity to *mcr-1* and was found in colistin-resistant isolates that did not contain *mcr-1*, and appeared to be mobilized on an IS1595 transposon[35]. Despite the different insertion sequences, intriguingly, this mobile element also contained a similar protein downstream of the *mcr* gene. Indeed, in *mcr-1*, *-2*, and *-3*, the *mcr* gene has a downstream open reading frame encoding, respectively, a putative PAP2 protein[12], a PAP2 membrane-associated lipid phosphatase[35], and a diacylglycerol kinase[36], all of which have transmembrane domains and are involved in the phosphatidic acid pathway[39,40]. Although the PAP2-like orf in *mcr-1* has been shown not to be required for colistin resistance[41], the presence of similar sequences downstream of other *mcr* genes implies some functional role, either in the formation of the mobile element and/or in its continued mobilization.

In summary, we assembled the largest data set to date of *mcr-1*-positive sequenced isolates through our own sequencing efforts combined with an exhaustive search of publicly available sequence databases including unassembled data sets from the SRA. Although this allowed us to obtain a truly global data set of 457 *mcr-1*-positive isolates covering 31 countries and five continents, we appreciate that the data is likely affected by complex sampling biases, with an over representation of samples from places with active surveillance and well-funded research communities. Equally, although we took advantage of the most sophisticated bioinformatics and phylogenetic tools currently available, the complex 'Russian doll' dynamics of the transposon, plasmids, and bacterial host limits our ability to reach strong inferences on some important aspects of the spread of *mcr-1*. Nevertheless, we believe our results highlight the potential for phylogenetic reconstruction of antimicrobial resistance elements at a global scale. We hope that future efforts relying on more sophisticated computational tools and even more extensive genetic sequence data will become part of the routine toolbox in infectious disease surveillance.

## Methods

**Compilation of genomic data set.** We blasted for *mcr-1* in all NCBI GenBank assemblies (as of 16th March 2017, $n = 90,759$) using a 98% identity cutoff. 195 records (0.21%; 121 assemblies, 73 complete plasmids, one complete chromosome) contained at least one contig with a full-length hit to *mcr-1* (1626 bases). We only included samples with a single copy of *mcr-1*. The only isolate with multiple copies was a previously published isolate with three chromosomal copies of *mcr-1* and seven copies of IS*Apl1*[42].

We searched a snapshot of all WGS bacterial raw read data sets in the SRA (December 2016), looking for samples containing mcr-1 by using a k-mer index ($k = 31$), which we had previously constructed[43]; software available at: https://github.com/phelimb/bigsi. A total of 184 data sets were found to contain at least 70% of the 31-mers in *mcr-1*. After removing duplicates (i.e., those with a draft assembly available) we could assemble contigs with *mcr-1* for 152 of these.

Our final data set comprised 457 isolates from six genera across 31 different countries, ranging in date from 2008 to 2017. Where only a year was provided as the date of isolate collection the date was set to the midpoint of that year.

Whenever identified isolates did not comprise previously assembled genomes or complete plasmids, the raw fastq files were first inspected using FastQC and trimmed and filtered on a case-by-case basis. De novo assembly was then conducted using Plasmid SPADES 3.10.0 using the *–careful* switch and otherwise

default parameters[44]. For those isolates sequenced using PacBio a different pipeline was employed. Correction, trimming, and assembly of raw reads was performed using Canu[45] and assembled reads were corrected and trimmed using the tool Circlator[46]. The quality of resultant assemblies was assessed using infoseq. In both cases the *mcr-1* carrying contigs in this final data set were identified using blastn v2.2.31[47].

We ran Plasmid Finder 1.3[48] with 95% identity to identify plasmid replicons on the *mcr-1* carrying contigs. In total, 182 unique contigs could be assigned a plasmid type using this method.

**Novel samples from China.** We selected 110 *mcr-1*-positive isolates from China for WGS from a larger survey effort of both clinical and livestock isolates. A total of 2824 non-repetitive clinical isolates, including 1637 *E. coli* and 1187 *K. pneumoniae* were collected from 15 provinces of mainland China from 2011 to 2016. Seventy-two isolates were resistant to polymyxin B, comprising 40 *E. coli* and four *K. pneumoniae* carrying *mcr-1*. Livestock samples were collected from four provinces of China in 2013 and 2016. One broiler farm of the Shandong province provided chicken anal swabs, liver, heart, and wastewater isolated in 2013. In 2016, samples including feces, wastewater, anal swabs, and internal organs of sick livestock were collected from swine farms, cattle farms, and broiler farms in four provinces (Jilin, Shandong, Henan, and Guangdong). A total of 601 *E. coli* and 126 *K. pneumoniae* were isolated, of which 167 (137 *E. coli* and 30 *K. pneumoniae*) were resistant to polymyxin B. We detected *mcr-1* in 135 *E. coli* and two *K. pneumoniae*, as well as in eight *E. coli* isolated from environmental samples, which were collected from influents and effluents of four tertiary care teaching hospitals.

All of the isolates were sent to the microbiology laboratory of Peking University People's Hospital and confirmed by routine biochemical tests, the Vitek system (bioMérieux, Hazelwood, MO, USA) and/or MALDI-TOF (Bruker Daltonics, Bremen, Germany). The minimal inhibitory concentrations (MICs) of polymyxin B was determined using the broth dilution method. The breakpoints of polymyxin B for *Enterobacteriaceae* were interpreted with the European Committee on Antimicrobial Susceptibility Testing (EUCAST, http://www.eucast.org/clinical_breakpoints) guidelines. Colistin-resistant isolates (MIC of ≥2 µg/ml) were screened for *mcr-1* by PCR and sequencing as described previously[49].

**Identification and alignment of mcr-1 transposon.** We searched for the *mcr-1* carrying transposon region across isolates by blasting for its major components: IS*Apl1* (*Actinobacillus pleuropneumoniae* reference sequence: EF407820), *mcr-1* (from *E. coli* plasmid pHNSHP45: KP347127.1), and short sequences representing the sequences immediately upstream and downstream of *mcr-1* (from KP347127.1) using blastn-short. Contiguous sequences containing *mcr-1* were aligned using Clustal Omega[50] and then manually curated and amended using jalview[51], resulting in a 3679 bp alignment containing the common ~2600 bp identified by Snesrud and colleagues[12]. The downstream copy of IS*Apl1* was more often fragmented or inverted. Twenty-eight isolates, which were all assemblies from the same study in Vietnam had a ~1.7 kb insertion downstream of *mcr-1* (Fig. 3e) before the downstream IS*Apl1* element.

**Phylogenetic analyses.** For constructing the transposon phylogeny, we excluded the downstream IS*Apl1* and the insertion sequence observed in a small number of samples, as well as regions identified as having signals of recombination by ClonalFrameML[52], resulting in a 3522 bp alignment. We removed two homoplastic sites (requiring >1 change on the phylogeny), before constructing a maximum parsimony neighbor-joining tree based on the Hamming distance between sequences. We calculated branch lengths using non-negative least squares with nnls.phylo in phangorn v2.2.0[53]. Phylogenies were visualized with ggtree v1.8.1[54].

**Phylogenetic dating.** Given recombination can conceal the clonal phylogenetic signal we also applied ClonalFrameML[52] to identify regions of high recombination in a subset of IncI2 and IncX4 plasmid background alignments. Where recombination hotspots were identified, they were removed from the alignment. In the IncI2 alignment, this resulted in removing 1281 positions. No regions of high recombination were detected in the IncX4 alignment. We applied root-to-tip correlations to test for a temporal signal in the data using TempEST[55]. There was a significantly positive slope for all three alignments (Supplementary Fig. 9–11).

We applied BEAUTi and BEAST v2.4.7[56,57] to estimate a timed phylogeny from an alignment of IncI2 plasmids (7161 sites, 110 isolates) and IncX4 plasmids (34,761 sites, 8 isolates). Sequences were annotated using their known sampling times expressed in years. For both plasmid alignments, the HKY substitution model was selected based on evaluation of all possible substitution models in bModelTest. BEAST analyses were then applied under both a coalescent population model (the coalescent Bayesian skyline implementation) and an exponential growth model (Coalescent Exponential population implementation). In addition, a strict clock, with a lognormal prior, and a relaxed clock (both lognormal and exponential) were tested. MCMC was run for 50,000,000 iterations sampling every 2000 steps and convergence was checked by inspecting the effective sample sizes and parameter value traces in the software Tracer v1.6.0. Analyses were repeated three times to ensure consistency between the obtained posterior distributions. Posterior trees for the best fitting model were combined in TreeAnnotator after a

10% burn-in to provide an annotated MCC tree. MCC trees were plotted using ggtree[54] for both backgrounds: IncI2 (Supplementary Fig. 12) and IncX4 (Supplementary Fig. 13). The model fit across analyses was compared using the Akaike's information criteria model through 100 bootstrap resamples as described in Baele and colleagues[58] and implemented in Tracer v1.6 (Supplementary Table 2).

Phylogenetic dating on the transposon was performed using an alignment of 364 isolates, which included only those with information on isolation date, across 3522 sites. As before BEAST analyses were applied under both a coalescent population model (coalescent Bayesian skyline implementation) and an exponential growth model (coalescent exponential population implementation). In addition, a strict clock, with a lognormal prior, and a relaxed clock (both lognormal and exponential) were tested. Analyses were run under a HKY substitution model for 600 million iterations sampling every 5000 steps. Only analyses using a strict clock model reached convergence after 600 million iterations. The resultant set of trees were thinned by sampling every 10 trees and excluding a 10% burn-in and combined using TreeAnnotator to produce a MCC tree. MCC trees were plotted using ggTree[48]. As before the model fit was evaluated using AICM's implemented in Tracer v1.6.

**Environmental distribution**. For the purpose of testing the distribution of sequences containing some trace of ISApl1, we classed isolates into broad categories as either environmental ($n = 39$; bird, cat, dog, fly, food, penguin, reptile, vegetables), agricultural ($n = 213$; chicken, cow, pig, poultry feed, sheep, turkey), or human ($n = 108$). We did not correct for study site with subsampling as we found great diversity within sites, consistent with a recent study showing multiple diverse mcr-1-positive strains within a single hospital sewage sample[59].

**Data access**. All the data generated or analyzed in this study are available within the paper and its supplementary information files. Accession numbers and metadata for all 457 isolates is provided in supplementary data 1. The newly sequenced 110 mcr-1-positive genomes have been submitted to the Short Read Archive under Bioproject: PRJNA408214, Accession: SRP118547.

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

## Acknowledgements

L.v.D., X.D., H.W., T.S., L.A.W., and FB acknowledge financial support from the Newton Trust UK-China NSFC initiative (grants MR/P007597/1 and 81661138006). F.B. additionally acknowledges support from the BBSRC GCRF scheme. L.P.S. was supported by a PhD scholarship from EPSRC (EP/F500351/1). P.B. is funded by Wellcome Trust on a "Genomic Medicine and Statistics DPhil" grant. A.R. was co-funded by the European Union: European regional development fund (ERDF), by the Conseil Régional de La Réunion and by the Centre de Coopération internationale en Recherche agronomique pour le Développement (CIRAD). H.W. was supported by National Natural Science Foundation of China (81625014). L.A.W. is supported by a Dorothy Hodgkin Fellowship funded by the Royal Society (Grant Number DH140195) and a Sir Henry Dale Fellowship jointly funded by the Wellcome Trust and the Royal Society (Grant Number 109385/Z/15/Z). Z.I. was funded by a Sir Henry Dale Fellowship jointly funded by the Wellcome Trust and the Royal Society (Grant Number 102541/A/13/Z). The funders had no role in study design, data collection and interpretation, or the decision to submit the work for publication.

## Author contributions

H.W. and F.B. conceived the project and designed the experiments. R.W., Q.W., X.W., L. J. Q.Z., Y.L., and H.W. collected samples. R.W., Q.W., X.W., LJ., Q.Z., and Y.L. performed microbial identification, antimicrobial susceptibility testing, screening for mcr-1, and DNA extraction for WGS. R.W. L.v.D., L.P.S. assembled the new sequence data. L.v. D. and L.P.S. curated the global data set and performed the computational analyses. T.D.-S. advised on functional aspects of colistin resistance. A.R., L.A.W., and X.D. helped with the phylogenetic reconstructions. P.B. and Z.I. performed the search for mcr-1-positive samples on the Short Read Archive. H.W. takes responsibility for the accuracy and availability of the epidemiological and raw sequence data, and F.B. for all bioinformatics and computational methods and results. L.v.D., L.P.S., and F.B. wrote the paper with contributions from X.D. and H.W. All authors read and commented on successive drafts and all approved the content of the final version.

## Additional information

**Competing interests:** The authors declare no competing financial interests.

