## [Peer Review File · Nature Communications]

Reviewers' comments:

Reviewer #1 (Remarks to the Author):

This manuscript by Wang et al. provides compelling arguments that the recent rise in *mcr-1* positive polymyxin resistant infections and colonization occurred through a single mobilization of *mcr-1* by an IS*Apl1* transposon. The authors analyzed all currently available whole genome sequences of *mcr-1* positive isolates, representing a large international and diverse isolate collection. They demonstrated that all *mcr-1* sequences contain a shared sequence that extends beyond the gene, suggesting the single mobilization, confirming previously postulated models. In addition, a large number of samples showed lack of IS*Apl1*, suggesting that the *mcr-1* transposon has been stabilized in a large proportion of isolates. Applying Bayesian reconstruction to this dataset they dated the mobilization of the transposon to 2006.

Overall this is a very nice study proposing a coherent recent evolutionary history and proposing a robust model of how this resistance gene may have spread rapidly over the past decade. Limitations of this study that need to be considered include a potential sampling and sequencing bias, which should be discussed in more detail.

Specific Comments:

1. Why were isolates with multiple copies excluded? How would inclusion of these influence the conclusions of the study?
2. Figure 3. Panel a lacks axis labeling (number of isolates?). Unable to read legend in Panel b.
3. Were short-read datasets reliable in ascertaining the structure of the putative *mcr-1* cassette and transposon in all isolates?
4. The plasmid background could not be ascertained in a large subset of isolates – please explain further. For these sequences, was there still sufficient confidence regarding the mapping of the *mcr-1* containing region.
5. The description of samples (n=457) should also include a breakdown of animal versus human. This is relevant for the section on environmental distribution. It appears that the total number of isolates included here was 360 rather than 457.
6. For the environmental comparisons did you account for study site? Samples from the same site (i.e. pig farm or hospital / household) are not independent and could bias the distribution of genotypes.
7. Line 72/73 – the text mentions spread of *mcr-1* within the hospital environment but the reference corresponds to Dutch travellers.

Reviewer #2 (Remarks to the Author):

Mcr-1 is the first transmissible gene responsible for resistance to colistin that has been characterized. It is of great public health importance, even if the level of resistance reached is generally lower than the one resulting from chromosomal mutations. The interest for this gene has been considerable with already an abundant literature. Major issues are the mechanism of mobilization of this gene as a composite transposon, its origin and its spreading. In the present article, Wang and al have addressed these different questions through the analysis of the sequences of 457 *mcr-1* loci (from genome and plasmid sequences). It included 347 sequences from the internet and 110 additional newly sequenced isolates from China.

The main results of this study were summarized in the first paragraph of the discussion:

- A single integration of *mcr-1* into an IS*Apl1* composite transposon.
- Its spread between multiple genomic backgrounds
- An "age" to the genesis of composite transposon

This manuscript is nicely written, but despite the extend of the analysis except the “age” of composite transposon the authors provided mostly an elegant confirmation of already published conclusions. A single integration into an ISAp11 composite transposon was determined in reference 12 (Snesrud, E. et al. A Model for Transposition of the Colistin Resistance Gene *mcr-1* by ISAp11. *Antimicrob. Agents Chemother.* 60, 6973–6976 (2016).) These authors clearly showed the uniqueness of this transposon and the subsequent loss of one or two ISAp11 leading to its stabilization. The spread between multiple genomics backgrounds was also clearly shown by Matamoros et al in reference 15: “Global Phylogenetic Analysis Of *Escherichia coli* And Plasmids Carrying The *mcr-1* Gene Indicates Bacterial Diversity But Plasmid Restriction” now recently published in *Scientific Reports*. To give an “age” to composite transposon is obviously useful. However, such recent emergence (in 2008) is not fully surprising and remains quite uncertain in particular given the occurrence of recombination.

Reviewer #3 (Remarks to the Author):

In the submitted manuscript, Wang, van Dorp and Shaw et al. employ a dataset of 457 existing as well as 110 newly sequenced isolates to bioinformatically investigate the genomic context and temporal origin of the colistin resistance gene *mcr-1*. While similar global demographic and genetic studies have been carried out in smaller scale, these authors are the first to estimate time of *mcr-1* mobilization by ISAp11.

The use of phylogenetic tools in resistance epidemiology is timely and important to understand the evolution of novel resistance types and the influence of genetic contexts herein. Although the focus on plasmids is well deserved, the manuscript would benefit from analysis on the broader genomic context in which these genotypes were sampled.

Major comments:

1.

The authors rightfully focus on plasmids as the most important locus of *mcr-1* spread and evolution. However, the host strain might also be important for the rate and mode of evolution which has been shown previously for multidrug resistance plasmids (e.g. 10.1093/molbev/msw163 and 10.1093/molbev/msv072). Therefore, it would strengthen the conclusions of the manuscript (and justify the final line of their abstract) to conduct subanalysis, similar to that of plasmid Inc-groups, on different species (even clone-types) to assess differential context evolution (similar to Fig. 3 b) or rates of evolution. It would be interesting to know if some strains support a more stable context than others, and whether the results in Fig. 3 b are biased by host-backgrounds.

2.

Similarly, *mcr-1* has also been found integrated into the chromosomes of some isolates. Did the authors include these in their analysis, and how do these related to the results obtained from the plasmid located genes?

3.

Line 296-298

The authors mention the detection of *mcr-1* positive *E. coli* isolates dating back to the 1980s. As *mcr-1* is unlikely to have evolved in *E. coli*, it would be of significant interest to obtain these isolates and assess the immediate (mobile) genetic context to show that this is different from the one dominating today.

The phrasing “...would be consistent with an early emergence but a long dormancy before the formation of the composite transposon” implies that the transposon was not present in these isolates, but this is not certain. The authors should at least underline this uncertainty e.g. in the (sampling)

bias of their model estimates.

Minor comments:

Line 72-73:

Reference 19 does not support spread within hospital environments. Please rephrase or update reference.

Line 113 and 123. Should be "the" Shandong province.

Figure 1:

The numbers are hard to read and some pie-charts overlap physically and in their colouring. Consider bigger numbers with a stronger colour and collapsing cramped pie-charts (e.g. Those in Europe) and to change the chronobacter/coli colour.

Figures 4 and 5:

The branch labels and legend (plasmid types, Figure 5) are hard to read and the colours used for Inc-groups are hard to distinguish.

Line 272:

Missing reference for "upstream copy being functionally more important."

Line 412-413:

Many plasmids contain multiple replicons, which makes plasmid typing at the contig level somewhat inaccurate. Did the authors consider this bias?

Response to reviewers

We would like to thank the Editor for the opportunity to submit a revision, and wish to thank the reviewers for the time and effort devoted to comment on our manuscript. We are grateful for the feedback we received and have followed essentially all suggestions, which we feel has been extremely useful for improving the manuscript.

A point-by-point response to all comments is included below. Our responses are shown in **red** immediately underneath the reviewers' comments. Where we reference line numbers, these refer to the resubmitted version of the manuscript.

We hope that with these changes the manuscript is now suitable for publication. Thank you for your time and consideration.

Thank you for your time and consideration,

Sincerely yours,

Lucy van Dorp, Liam Shaw and Francois Balloux, on behalf of all the co-authors

Reviewer #1 (Remarks to the Author):

This manuscript by Wang et al. provides compelling arguments that the recent rise in *mcr-1* positive polymyxin resistant infections and colonization occurred through a single mobilization of *mcr-1* by an IS*Apl1* transposon. The authors analyzed all currently available whole genome sequences of *mcr-1* positive isolates, representing a large international and diverse isolate collection. They demonstrated that all *mcr-1* sequences contain a shared sequence that extends beyond the gene, suggesting the single mobilization, confirming previously postulated models. In addition, a large number of samples showed lack of IS*Apl1*, suggesting that the *mcr-1* transposon has been stabilized in a large proportion of isolates. Applying Bayesian reconstruction to this dataset they dated the mobilization of the transposon to 2006.

Overall this is a very nice study proposing a coherent recent evolutionary history and proposing a robust model of how this resistance gene may have spread rapidly over the past decade. Limitations of this study that need to be considered include a potential sampling and sequencing bias, which should be discussed in more detail.

Specific Comments:

1. Why were isolates with multiple copies excluded? How would inclusion of these influence the conclusions of the study?

Only one isolate identified in our initial dataset appeared to have multiple copies: NCBI accession CP016182, which had 3 chromosomal copies of *mcr-*

1 and 7 copies of *ISAp11*. This corresponds to *E. coli* isolated EC590 with a tandem arrangement: *ISAp11-mcr-1-Δpap-ISAp11-mcr-1-Δpap-ISAp11-mcr-1-Δpap-ISAp11*. (Yu et al. 2017, <https://doi.org/10.1093/jac/dkw541>). We confirmed this arrangement and provide a table of the positions of the relevant elements (Table 1, below). We aligned the regions 4224432-4228110 (r1), 4228111-4231769 (r2), and 4231790-4236538 (r3). These regions were identical except for at position 80 in the alignment (r1: T, r2/r3: C), supporting the conclusion of Yu et al. that this represented a triplication of the locus, although the mechanism remains unknown. Including this isolate would not therefore have influenced the conclusions of the study. This single isolate with multiple copies is now mentioned in the Methods section together with the initial reference (lines 355-356).

Table 1: positions of relevant elements of the composite transposon in CP016182, which contains 3 copies of *mcr-1*.

	Start	End	Length
ISAp11	4224432	4225501	1070
mcr-1	4225688	4227313	1626
Δpap	4227385	4228107	723
ISAp11	4228111	4229180	1070
mcr-1	4229367	4230992	1626
Δpap	4231064	4231786	723
ISAp11	4231790	4232859	1070
mcr-1	4233046	4234671	1626
Δpap	4234743	4235465	723
ISAp11	4235469	4236538	1070

2. Figure 3. Panel a lacks axis labeling (number of isolates?). Unable to read legend in Panel b.

We have added the axis labelling to panel a and increased the font size for the legend in panel b as well as adding axis labels.

3. Were short-read datasets reliable in ascertaining the structure of the putative *mcr-1* cassette and transposon in all isolates?

We believe so. The start of the composite transposon alignment was on average quite distant from the start of the contig (mean: 20,341 bases) so for the majority of sequences the whole alignment was well-determined in its genomic context. There were 122 sequences where the alignment started at the start of the contig i.e. we could not ascertain the upstream genomic context with short read sequencing. Nevertheless, we do not see this as a problem for the transposon alignment, because the structure could still reliably be determined in all isolates.

4. The plasmid background could not be ascertained in a large subset of isolates – please explain further. For these sequences, was there still sufficient confidence regarding the mapping of the *mcr-1* containing region.

We were unable to assign plasmid types to plasmids which lacked a replicon sequence characterised in the PlasmidFinder reference dataset. As a consequence, we had more success assigning types to complete plasmids (62/73 assigned) and those of longer contig length (52450bp (32870-63490) assigned versus 42610bp (2928-22170) unassigned; mean (5-95% CI)). Mapping of the *mcr-1* containing regions in different plasmid backgrounds was ascertained based on the alignment and thus independent of our ability to assign plasmid type.

5. The description of samples (n=457) should also include a breakdown of animal versus human. This is relevant for the section on environmental distribution. It appears that the total number of isolates included here was 360 rather than 457.

We were missing metadata information on the host for n=97 isolates, hence the reduced n=(457-97)=360 for this analysis. We have added a line to the description of samples in the results to give the animal vs. human breakdown (lines 91-93) as we agree this is of interest:

Isolates with metadata on the sample source (n=360) came from a range of animal (n=222), human (n=108) and environmental (n=30) hosts.

We have now also included a new supplementary figure (Figure S2) with the tips of the transposon phylogeny annotated for sample source, similarly to figures 4 and 5.

6. For the environmental comparisons did you account for study site? Samples from the same site (i.e. pig farm or hospital / household) are not independent and could bias the distribution of genotypes.

We did not account for study site and have emphasized this in the text (lines 486-489):

We did not correct for study site with subsampling as we found great diversity within sites, consistent with a recent study showing multiple diverse mcr-1 positive strains within a single hospital sewage sample⁵⁸.

7. Line 72/73 – the text mentions spread of mcr-1 within the hospital environment but the reference corresponds to Dutch travellers.

Apologies, this should have instead been a reference to Tian et al. (2017) 'MCR-1-producing *Klebsiella pneumoniae* outbreak in China'. Corrected.

Reviewer #2 (Remarks to the Author):

Mcr-1 is the first transmissible gene responsible for resistance to colistin that has been characterized. It is of great public health importance, even if the level of resistance reached is generally lower than the one resulting from chromosomal mutations. The interest for this gene has been considerable with already an abundant literature. Major issues are the mechanism of mobilization of this gene as a composite transposon, its origin and its spreading. In the present article, Wang and al have addressed these different questions through the analysis of the sequences of 457 mcr-1 loci (from

genome and plasmid sequences). It included 347 sequences from the internet and 110 additional newly sequenced isolates from China.

The main results of this study were summarized in the first paragraph of the discussion:

- A single integration of *mcr-1* into an IS*Apl1* composite transposon.
- Its spread between multiple genomic backgrounds
- An “age” to the genesis of composite transposon

This manuscript is nicely written, but despite the extend of the analysis except the “age” of composite transposon the authors provided mostly an elegant confirmation of already published conclusions.

A single integration into an IS*Apl1* composite transposon was determined in reference 12 (Snesrud, E. et al. A Model for Transposition of the Colistin Resistance Gene *mcr-1* by IS*Apl1*. *Antimicrob. Agents Chemother.* 60, 6973–6976 (2016).) These authors clearly showed the uniqueness of this transposon and the subsequent loss of one or two IS*Apl1* leading to its stabilization.

We agree that the single integration into an IS*Apl1* composite transposon was suggested by Snesrud et al. and their invaluable paper greatly helped us to produce the transposon alignment. However, we confirmed their evolutionary model on a much larger dataset, and managed to date the insertion event.

The spread between multiple genomics backgrounds was also clearly shown by Matamoros et al in reference 15: “Global Phylogenetic Analysis Of *Escherichia coli* And Plasmids Carrying The *mcr-1* Gene Indicates Bacterial Diversity But Plasmid Restriction” now recently published in *Scientific Reports*.

We were very pleased to see this published after we had submitted, as we had cited the biorxiv preprint. We have updated this reference to the complete paper, and made it clear that the plasmid diversity we observe is consistent with their findings (lines 193-195):

Incl2 and *IncX4* were the dominant plasmid types, accounting for 47% and 36% of the isolates, respectively (Figure 5) similar to the proportions observed by Matamoros et al.¹⁵.

To give an “age” to composite transposon is obviously useful. However, such recent emergence (in 2008) is not fully surprising and remains quite uncertain in particular given the occurrence of recombination.

The date we obtain must necessarily pre-date 2008 due to the presence of an isolate from 2008 in our dataset. We were personally surprised by the recent emergence, in particular given reports of earlier *mcr-1* positive samples. We are confident our results were not coloured by recombination, as we removed all sites putatively affected by genetic recombination.

Reviewer #3 (Remarks to the Author):

In the submitted manuscript, Wang, van Dorp and Shaw et al. employ a dataset of 457 existing as well as 110 newly sequenced isolates to bioinformatically investigate the genomic context and temporal origin of the

colistin resistance gene *mcr-1*. While similar global demographic and genetic studies have been carried out in smaller scale, these authors are the first to estimate time of *mcr-1* mobilization by ISAp11.

The use of phylogenetic tools in resistance epidemiology is timely and important to understand the evolution of novel resistance types and the influence of genetic contexts herein. Although the focus on plasmids is well deserved, the manuscript would benefit from analysis on the broader genomic context in which these genotypes were sampled.

Major comments:

1. The authors rightfully focus on plasmids as the most important locus of *mcr-1* spread and evolution. However, the host strain might also be important for the rate and mode of evolution which has been shown previously for multidrug resistance plasmids (e.g. 10.1093/molbev/msw163 and 10.1093/molbev/msv072). Therefore, it would strengthen the conclusions of the manuscript (and justify the final line of their abstract) to conduct subanalysis, similar to that of plasmid Inc-groups, on different species (even clone-types) to assess differential context evolution (similar to Fig. 3 b) or rates of evolution. It would be interesting to know if some strains support a more stable context than others, and whether the results in Fig. 3 b are biased by host-backgrounds.

We thank the reviewer for this suggestion, and certainly agree this is an interesting point. However, the sample is not well suited to explore the possible role of host species in the spread of the *mcr-1* element, due to the dominance of *E. coli* (n=411, 89.9% of samples). Nevertheless, we think it is right that some comment on the possible different context for evolution of the composite transposon is important. We have therefore produced two new supplementary figures annotating the transposon phylogeny by bacterial species and sample source (animal/environmental/human) and added the following sentence to the text (lines 172-174):

There was no discernible clustering of isolates by bacterial species (Figure S1) or sample source (Figure S2), suggesting the composite transposon does not evolve differently in these different backgrounds.

2. Similarly, *mcr-1* has also been found integrated into the chromosomes of some isolates. Did the authors include these in their analysis, and how do these related to the results obtained from the plasmid located genes?

We identified *mcr-1* on one closed chromosome. We cannot formally rule out there may be a few other chromosomal *mcr-1*-containing contigs. This has now been stated in the text (lines 195-198):

*One isolate in our dataset was definitively located on a complete chromosome. Though, we cannot rule out the presence of a few other chromosomal copies of *mcr-1* located on short contigs.*

3. Line 296-298

The authors mention the detection of *mcr-1* positive *E. coli* isolates dating back to the 1980s. As *mcr-1* is unlikely to have evolved in *E. coli*, it would be

of significant interest to obtain these isolates and assess the immediate (mobile) genetic context to show that this is different from the one dominating today.

We would also be very interested to perform this analysis, but do not have access to these samples. We hope the authors of this isolated observation may be prompted to do so by our paper!

The phrasing "...would be consistent with an early emergence but a long dormancy before the formation of the composite transposon" implies that the transposon was not present in these isolates, but this is not certain. The authors should at least underline this uncertainty e.g. in the (sampling) bias of their model estimates.

We thank the reviewer for pointing out that this section was confusing. We acknowledge that this observation is difficult to reconcile with our results and are now explicit about the fact that this early observation of *mcr-1* is not compatible with our inferred date for the integration of *mcr-1* in the ISAp11 transposon. The text in the discussion has been shortened and clarified (lines 272-274):

This observation seems surprising in light of our results, which clearly exclude such an early spread of mcr-1 at least on this ISAp11 transposon background.

Minor comments:

Line 72-73: Reference 19 does not support spread within hospital environments. Please rephrase or update reference.

Apologies, this should have instead been a reference to Tian *et al.* (2017) 'MCR-1-producing *Klebsiella pneumoniae* outbreak in China'. Corrected.

Line 113 and 123. Should be "the" Shandong province.

Corrected.

Figure 1: The numbers are hard to read and some pie-charts overlap physically and in their colouring. Consider bigger numbers with a stronger colour and collapsing cramped pie-charts (e.g. Those in Europe) and to change the chronobacter/coli colour.

Corrected.

Figures 4 and 5: The branch labels and legend (plasmid types, Figure 5) are hard to read and the colours used for Inc-groups are hard to distinguish.

We appreciate this point. We have increased the text size and reformatted the colours for the Inc groups, and hope this makes it more legible.

Line 272: Missing reference for "upstream copy being functionally more important."

We have added this reference to the main text (Snesrud *et al.* 2016).

Line 412-413: Many plasmids contain multiple replicons, which makes plasmid typing at the contig level somewhat inaccurate. Did the authors consider this bias?

We agree that plasmid-typing at the contig level has this limitation. Of our 457 strong dataset, we assigned 182 unique plasmid types of which 62 were from complete plasmids. Of the remainder: 60 Assemblies, 22 Illumina WGS, 11 NextSeq 500, 8 Illumina MiSeq, 7 HiSeq X Ten, 5 unspecified, 3 Pacbio, 3 Illumina HiSeq 2000 and 1 Illumina HiSeq 2500. As one might expect, contigs which comprised plasmid replicons tended to be longer: the mean contig length where a plasmid type was assigned was 52,450bp (5-95% CI: 32,870-63,490) while those unassigned had a mean contig length of 42,610bp (5-95% CI: 2928-22,170). 9 contigs were assigned to more than one plasmid type consistent with the presence of multiple unique plasmid replicons. These assignments were excluded from the final analyses.

REVIEWERS' COMMENTS:

Reviewer #1 (Remarks to the Author):

No further comments

Reviewer #3 (Remarks to the Author):

I think my comments have been adequately addressed.